# CLAD: CONTINUAL LEARNING FOR ROBUST ADVERSARIAL TEXT DETECTION AND REPAIR IN RESOURCE-CONSTRAINED SCENARIOS

## ABSTRACT

Textual adversarial attacks present a critical threat to NLP systems by subtly altering inputs to deceive models, necessitating robust detection and defense mechanisms. Traditional methods, however, suffer from high computational costs, poor generalization to unseen attacks, and vulnerability to distribution shifts, particularly in resource-constrained scenarios where adversarial example sampling is expensive and scarce. To address these challenges, we propose CLAD, a continual learning-based framework for adversarial detection and repair, designed to enhance robustness and transferability in low-resource environments. By leveraging continual learning, CLAD mitigates catastrophic forgetting of learned adversarial patterns and incrementally improves generalization as new attack types are introduced. CLAD integrates two adversarial repair methods that preserve semantic fidelity while neutralizing perturbations. Across four text classification datasets and three primary attacks (BAE, PWWS, TextFooler), CLAD improves with larger memory buffers (MS $\in \{0, 1, 10, 100\}$) and exhibits reduced forgetting. The best *detection* accuracy reaches **82.20%** (Amazon, in-domain, MS=100), while on the same dataset *defense* achieves up to **99.65%** defense accuracy (D.A.) and **84.73%** recovery accuracy (R.A.) against TextFooler via $PD_{LLM}$.

## 1 INTRODUCTION

Textual adversarial attacks pose a growing and critical threat to natural language processing (NLP) models, particularly pretrained language models (PLMs), by subtly modifying input texts in ways that are imperceptible to humans but can deceive classifiers or other downstream components, ultimately leading to severe performance degradation. For instance, early studies Li et al. (2019); Ebrahimi et al. (2018) primarily exploited character-level perturbations (e.g., "GOOD" $\rightarrow$ "GO0D") to manipulate lexical or statistical patterns that models rely on Ebrahimi et al. (2018); Li et al. (2019). Neural systems have been shown to be particularly vulnerable to such attacks, raising serious concerns about the reliability and security of modern NLP pipelines.

In response to these challenges, adversarial defense methods have been developed to detect and mitigate malicious inputs. Adversarial detection aims to identify whether a given input is adversarial, while adversarial defense focuses on repairing such inputs to recover correct predictions. However, the evolution of defense strategies has lagged behind the increasing diversity of textual attacks. Moreover, existing defense approaches are often computationally expensive, as they typically operate in a non-targeted manner, requiring the generation of multiple plausible candidates to ensure effectiveness, especially for voting- or reconstruction-based methods Wang et al. (2022b); Mozes et al. (2021); Swenor & Kalita (2022).

Recent studies suggest that the detect-to-defend Bao et al. (2021); Zhou et al. (2019) paradigm can reduce unnecessary overhead by selectively defending only inputs identified as adversarial, provided that the detector has been trained on a sufficiently large and diverse set of adversarial examples. Nonetheless, this paradigm still incurs significant computational cost during defense, due to steps such as adversarial augmentation and ensemble-based prediction Dong et al. (2021b). As a result, most current adversarial detection and defense pipelines rely heavily on large-scale training data and computationally intensive repair strategies. These issues are particularly pronounced in low-resource

settings, where access to adversarial examples is limited and computational budgets are constrained. Furthermore, inaccurate adversarial detection can worsen model performance Shen et al. (2023), especially when incorrect assumptions lead to faulty repairs that introduce new vulnerabilities.

Compounding the issue is the well-documented vulnerability of adversarial defenses to distributional shifts. Even with advanced text augmentation techniques, detection mechanisms trained on one data distribution often fail to generalize to unseen or evolving domains. This problem is exacerbated in low-resource environments, where collecting diverse adversarial examples is costly and time-consuming. Consequently, detectors trained on narrow attack distributions frequently struggle to transfer knowledge to novel attack types or domain shifts. As shown in Table 1, models experience substantial performance degradation, up to 12.91% forgetting rate in cross-domain SST2 adversarial detection, when exposed to underrepresented or entirely novel adversarial patterns. Alarmingly, misclassifications in such settings can propagate through downstream components, triggering a cascade of poor decisions and compounding performance loss.

To address these limitations, we propose a continual learning (CL)-based paradigm for adversarial detection and defense, tailored for resource-constrained environments. Continual learning offers two key advantages: (1) it mitigates catastrophic forgetting Kirkpatrick et al. (2017); Aljundi et al. (2018), where prior knowledge erodes as new information is introduced, and (2) it supports incremental learning, which is especially beneficial when only a small number of adversarial examples are available at a time Biesialska et al. (2020); De Lange et al. (2021). By balancing knowledge retention and adaptation to new adversarial patterns, continual learning enables a resilient and evolving defense mechanism that remains effective over time. Moreover, the incremental incorporation of adversarial samples reduces the need for large upfront datasets, thereby lowering computational overhead Buzzega et al. (2020). This makes CL especially attractive for real-world deployment scenarios involving dynamic data distributions and constrained resources.

Building on these insights, we introduce a novel framework, CLAD, which systematically integrates continual learning into the adversarial detection and repair pipeline following the detect-to-defend paradigm. We conduct comprehensive evaluations of CLAD across multiple NLP datasets, attack strategies, and PLMs, with a particular focus on both detection accuracy and downstream task stability in low-resource settings. Experimental results demonstrate that continual learning significantly enhances the generalization and robustness of adversarial detectors. These results underscore the potential of continual learning as a lightweight and computationally efficient solution for addressing evolving adversarial threats, particularly in edge environments.

Our main contributions are as follows:

- Framework Design: We propose a continual learning-based adversarial detection and repair framework tailored for resource-constrained settings. Our method outperforms baseline approaches, achieving up to **10.63%** higher detection accuracy and **68.93%** better defense recovery performance across four datasets and three adversarial attack types.

- Continual Learning Analysis: We conduct an in-depth analysis of performance trajectories under different memory buffer sizes. Our experiments show that detection accuracy significantly benefits from increased memory capacity, although the gains plateau once the buffer size exceeds 10.

- LLM-based Repair Strategy: We introduce a large language model (LLM)-based adversarial repair strategy that effectively neutralizes perturbations while preserving semantic fidelity. This method outperforms traditional techniques like perturbation defocusing, which often produce semantically corrupted outputs Yang & Li (2024).

## 2 METHOD

In this section, we present CLAD, a framework designed for adversarial *detection* and *defense* in resource-constrained environments. CLAD is compatible with pre-trained language models (PLMs) and is capable of accurately identifying and repairing adversarial samples, even when faced with sophisticated attack methods. This capability enhances the overall performance and robustness of the models. Our approach comprises two primary components: (i) a standalone adversarial detector to identify malicious inputs and (ii) an adversarial defense module to repair them, restoring the model's

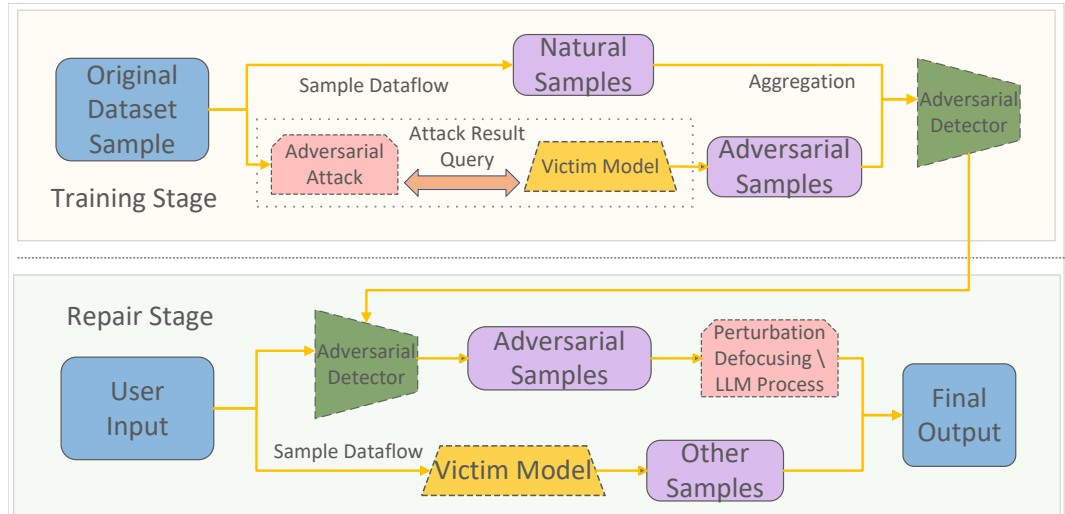

Figure 1: Workflow of the adversarial detection and defense framework (CLAD). The upper section illustrates the adversarial *detector* training process. Natural examples are sampled and passed to an adversarial attack module, which queries the victim model to generate adversarial examples. These successful adversarial examples are collected and aggregated with the natural ones to train a standalone adversarial detector. The lower section shows the deployment phase. User input is processed by the adversarial detector, which determines if the input is adversarial. If deemed adversarial, perturbation defocusing strategies are applied to repair the input, with the goal of restoring the victim model's correct prediction. This two-stage framework leverages adversarial example generation and continual learning to provide robust, detector-triggered defense capabilities for pre-trained language models.

original performance. We also integrate continual learning techniques to enhance adversarial detection in settings where attack patterns evolve. By leveraging continual learning, our framework incrementally adapts to new adversarial threats without compromising previously acquired knowledge, thereby ensuring sustained robustness and efficiency. The overall workflow of CLAD is depicted in Figure 1.

## 2.1 ADVERSARIAL DETECTION

The first component of our framework is a standalone adversarial detector. We detail the data sampling and training process below.

### 2.1.1 ADVERSARIAL EXAMPLE SAMPLING

The process begins by training an adversarial detector using a collection of natural and pre-sampled adversarial examples. To ensure diversity and computational feasibility, we adopt a *stratified adversarial sampling* strategy to construct our training dataset $\overline{\mathcal{D}}$:

$$\overline{\mathcal{D}} = \mathcal{D}_{\text{natural}} \cup \bigcup_{a \in \mathcal{A}} \mathcal{D}_a^{\text{adv}}, \tag{1}$$

where $\mathcal{D}_{\text{natural}}$ denotes the set of natural examples, and $\mathcal{D}_a^{\text{adv}}$ represents *successful* adversarial examples generated by a specific attack method $a$ from a set of attackers $\mathcal{A}$ (e.g., BAE, PWWS).

For each natural example $\langle \mathbf{x}, y \rangle \in \mathcal{D}_{\text{natural}}$, we generate adversarial candidates $\hat{\mathbf{x}}$ using an attacker $a \in \mathcal{A}$:

$$\hat{\mathbf{x}} \leftarrow a\big(F_{\text{victim}}, \mathbf{x}, y\big), \quad \text{retaining only if } F_{\text{victim}}(\hat{\mathbf{x}}) \neq y. \tag{2}$$

We sample up to $N_{\text{adv}} = 1000$ successful adversarial examples per dataset-attacker pair, balancing:

- **Attack Diversity**: Diverse representations of word-level perturbations.

- **Computational Cost**: A manageable number of total adversarial examples.

This strategy ensures broad coverage of perturbation types while maintaining computational tractability.

### 2.1.2 ADVERSARIAL DETECTOR TRAINING

Using the collected examples, we train a standalone adversarial detector. Inspired by data quality selection in LLM research Gunter et al. (2024); Dubey et al. (2024), we fine-tune a BERT model to distinguish between natural and adversarial inputs. Designing the detector as a separate module allows for flexible deployment across different victim models and facilitates continual learning without altering the victim model's parameters.

**Architecture** The detector $D_\theta$ consists of a BERT encoder followed by a binary classification head. Given an input $\mathbf{x}$:

$$h = \text{BERT}(\mathbf{x}), \quad z = W_{\text{det}}h_{[\text{CLS}]} + b_{\text{det}}, \quad p_{\text{adv}} = \sigma\left(z\right), \tag{3}$$

where $h_{[\text{CLS}]}$ is the final hidden state of the '[CLS]' token, $W_{\text{det}} \in \mathbb{R}^{1 \times d}$ and $b_{\text{det}} \in \mathbb{R}$ form a learnable projection layer, and $p_{\text{adv}}$ is the predicted probability that $\mathbf{x}$ is adversarial.

**Class-Imbalanced Optimization** To address the class imbalance between natural and adversarial examples (approx. 1:10), we employ two techniques:

- **Balanced Batch Sampling**: Each mini-batch is constructed with a 1:1 ratio of natural to adversarial examples.
- **Focal Loss** Li et al. (2020b): To focus training on harder-to-classify examples, we use the focal loss, defined as:

$$\mathcal{L}_{\text{det}} = -\alpha_t(1 - p_t)^\gamma \log p_t,$$
$$p_t = \begin{cases} p_{\text{adv}}, & \text{if } y^{\text{det}} = 1, \\ 1 - p_{\text{adv}}, & \text{if } y^{\text{det}} = 0. \end{cases} \tag{4}$$

where $y^{\text{det}} \in \{0, 1\}$ is the detection label (1 for adversarial), $\gamma = 2$ is the focusing parameter, and $\alpha_t$ dynamically balances class frequencies.

**Deployment** Once trained, the detector classifies an input $\mathbf{x}$ based on its predicted probability $p_{\text{adv}}$ and a predefined threshold $\tau \in (0, 1)$:

$$\text{Detector} : \mathbf{x} \mapsto \mathbb{I}[p_{\text{adv}}(\mathbf{x}) \geq \tau], \tag{5}$$

where the output is 1 if the input is deemed adversarial and 0 otherwise. The choice of $\tau$ allows for controlling the precision-recall trade-off for triggering the defense mechanism.

### 2.2 CONTINUAL LEARNING FOR ADVERSARIAL DETECTION

Adversarial threats are not static; attack strategies evolve, and data domains shift over time. A detector trained on one set of attacks may become obsolete as new threats emerge. Continual Learning (CL) provides a paradigm for this problem by enabling a model to adapt to a sequence of tasks $\{\mathcal{T}_1, \ldots, \mathcal{T}_T\}$ while mitigating catastrophic forgetting. A general CL objective for our detector can be formulated as:

$$\theta_t^* = \arg\min_\theta \underbrace{\mathbb{E}_{(\mathbf{x}, y^{\text{det}}) \sim \mathcal{D}_t}\left[\ell(D_\theta(\mathbf{x}), y^{\text{det}})\right]}_{\text{current-task adaptation}}$$
$$+ \lambda \underbrace{\sum_{k=1}^{t-1} \mathbb{E}_{(\mathbf{x}, y^{\text{det}}) \sim \mathcal{M}_k}\left[\ell(D_\theta(\mathbf{x}), y^{\text{det}})\right]}_{\text{past-knowledge consolidation}}, \tag{6}$$

where learning on the current data distribution $\mathcal{D}_t$ is regularized by replaying samples from a memory buffer $\mathcal{M}_k$ of past data.

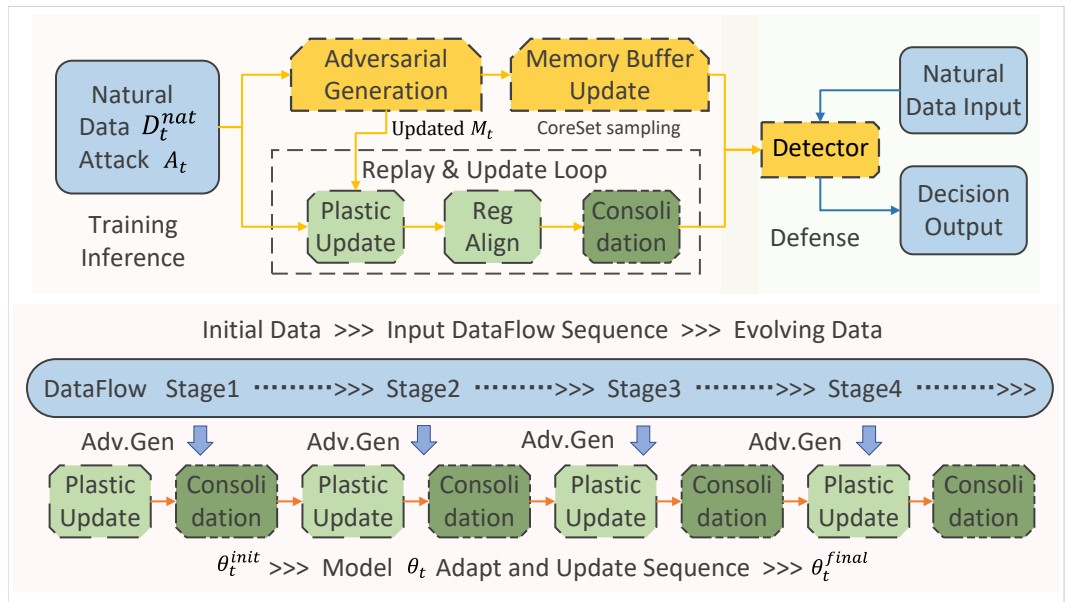

Figure 2: The upper panel illustrates the training pipeline within each stage: natural data is used to generate adversarial examples, which are stored in a dual-buffer memory (natural/adv). A two-phase update follows—(1) a Plastic Update leveraging current data to adapt to new threats, and (2) a Consolidation Update using replayed samples to preserve robustness to prior attacks. The lower panel shows the stage-wise model evolution across sequential attack scenarios, where continual updates help the detector maintain performance under evolving adversarial landscapes.

However, standard CL strategies are often insufficient for the adversarial setting. Generic replay methods treat all data equally, failing to prioritize critical new adversarial patterns. Regularization-based methods (e.g., EWC Kirkpatrick et al. (2017)) focus on protecting model parameters, which is less suited for a problem where the core challenge is adapting to a growing set of diverse attack types. Furthermore, such methods can introduce significant computational overhead, conflicting with our goal of a lightweight solution. As a proof-of-concept, we aim to find a minimal, off-the-shelf replay mechanism that maintains detection performance under evolving attacks without inflating complexity. An overview of our CL pipeline is presented in Figure 2.

### 2.2.1 EVOLUTION-AWARE ADVERSARIAL CONTINUAL LEARNING

We consider an evolving adversarial detection task where data arrives in stages $\{\mathcal{S}_1, \mathcal{S}_2, ..., \mathcal{S}_T\}$. Each stage $\mathcal{S}_t = (\mathcal{D}_t^{\text{nat}}, \mathcal{A}_t)$ contains natural samples and a new attack strategy. The detector must adapt to new attacks $\mathcal{A}_t$ while preserving its ability to detect all historical attacks, under a bounded memory budget.

**CoreSet Selection**  To manage the memory buffer, we use CoreSet selection. The operator $\text{CoreSet}(S, m)$ returns a subset $C \subseteq S$ of size $\lfloor m \rfloor$. This subset is selected by greedy $k$-center in the detector's representation space, which iteratively chooses the point that maximizes its minimum distance to the points already in the core set. This ensures a diverse summary of past data under a fixed budget.

### 2.3 TEXTUAL ADVERSARIAL DEFENSE

CLAD implements defense mechanisms that are triggered by the adversarial detector, a strategy known as *reactive adversarial defense*. Upon identifying an adversarial example $\hat{\mathbf{x}}$, CLAD caches the victim model's erroneous prediction, which we term the *fake prediction* ($\hat{y} = F_{\text{victim}}(\hat{\mathbf{x}})$). This cached prediction provides a crucial signal for the defense process.

Subsequently, CLAD engages in a *guided adversarial defense (repair)* process. The primary objective is to modify the detected adversarial example $\hat{\mathbf{x}}$ into a new version $\tilde{\mathbf{x}}^r$ such that the victim model's prediction on the repaired text is no longer the fake prediction (i.e., $F_{\text{victim}}(\tilde{\mathbf{x}}^r) \neq \hat{y}$). This *escape criterion* serves as a proxy for successful defense, especially in online settings where the ground-truth label is unknown. By using the cached fake prediction as weak supervision, we can mitigate malicious perturbations more effectively than defense methods that lack this guidance.

CLAD adopts two defense strategies that *defocus* the model from adversarial perturbations: **Paraphrase Defocusing** and **Perturbation Defocusing** Yang & Li (2024).

### 2.3.1 PARAPHRASE DEFOCUSING

Paraphrase defocusing leverages large language models (LLMs) to rephrase text while retaining semantic meaning. The core idea is that subtle alterations in phrasing can nullify malicious perturbations. We use an LLM (e.g., ChatGPT) to iteratively re-express a detected adversarial instance until the repaired text satisfies the escape criterion. The full loop is formalized as:

$$\tilde{\mathbf{x}}^r \leftarrow \text{PD}_{\text{LLM}}\Big(F_{\text{victim}}, \langle \hat{\mathbf{x}}, \hat{y} \rangle\Big), \tag{7}$$

where $\text{PD}_{\text{LLM}}$ is the iterative paraphrasing process detailed in Algorithm 2. This loop continues until the victim model's prediction differs from the cached fake prediction or a predefined iteration limit ($I_{\max}$) is reached.

### 2.3.2 PERTURBATION DEFOCUSING

Perturbation defocusing reverses the effects of malicious edits by repurposing an adversarial attacker as a controlled editor. It injects benign perturbations to steer the model away from the fake prediction. Given an adversarial input $\hat{\mathbf{x}}$ and its fake prediction $\hat{y}$, the process is:

$$\tilde{\mathbf{x}}^r \leftarrow \text{PD}_{\hat{\mathcal{A}}}\Big(F_{\text{victim}}, \langle \hat{\mathbf{x}}, \hat{y} \rangle\Big), \tag{8}$$

where $\text{PD}_{\hat{\mathcal{A}}}$ represents the perturbation defocusing process utilizing a chosen attacker $\hat{\mathcal{A}}$ (e.g., PWWS) as an editor.

As detailed in Algorithm 3, we iteratively introduce minimal benign changes via $\hat{\mathcal{A}}$ until the victim model's prediction deviates from $\hat{y}$ or the attacker fails to provide further valid perturbations. Because it operates independently of the victim model's parameters, this method is flexible and effective across diverse attack scenarios.

## 3 EXPERIMENTS

In this section, we comprehensively evaluate CLAD, our proposed framework for adversarial detection and defense in low-resource environments. Our experiments are designed to assess the effectiveness of adversarial detection, the robustness of adversarial defense mechanisms, and the adaptability of our framework through continual learning. We utilize multiple datasets, diverse adversarial attack methods, and state-of-the-art baseline defenses to ensure a thorough evaluation. The detailed experimental settings and workflow are described in Appendix E.

### 3.1 ADVERSARIAL DETECTION PERFORMANCE

We first evaluate the continual learning-based adversarial detector. Table 1 summarizes the detection accuracy (Acc) and forgetting rate (FR) across four datasets under both in-domain and cross-domain settings. The results show that increasing the memory buffer size (MS) generally improves detection accuracy and reduces forgetting. For instance, on AGNews in the in-domain setting, accuracy increases from 78.24% to 80.54% and the forgetting rate drops from 2.24 to $-1.97$ as MS grows from 0 to 100. In most cases, in-domain training outperforms cross-domain training, highlighting the importance of domain-specific adversarial examples; however, we also observe a counterexample on Yahoo!, where cross-domain training yields higher accuracy at all MS values. Notably, the Amazon dataset shows negative forgetting rates (e.g., $-3.20$ at MS=100), suggesting that continual exposure

to diverse attacks enhances the model's plasticity and can improve performance on previously seen tasks. A discussion concerning the negative forgetting rate is provided in the experimental section of the Appendix. Detailed performance breakdowns for each attack method are available in Figure 3 in the appendix.

| Dataset | In/Out Domain | MS=0 | | MS=1 | | MS=10 | | MS=100 | |
|---------|---------------|------|------|------|------|------|------|------|------|
| | | Acc ↑ | FR ↓ | Acc ↑ | FR ↓ | Acc ↑ | FR ↓ | Acc ↑ | FR ↓ |
| SST2 | In-Domain | 77.18 | 5.64 | 77.24 | 4.64 | 77.49 | 4.61 | 77.60 | 4.57 |
| | cross-Domain | 73.38 | 12.91 | 73.44 | 12.15 | 74.52 | 10.85 | 75.83 | 8.30 |
| Amazon | In-Domain | 80.93 | -1.39 | 81.21 | -1.31 | 80.77 | -0.68 | 82.20 | -3.20 |
| | cross-Domain | 77.11 | 5.42 | 78.40 | 4.65 | 77.97 | 4.66 | 79.36 | 2.34 |
| AGNews | In-Domain | 78.24 | 2.24 | 79.28 | -0.67 | 79.14 | -0.51 | 80.54 | -1.97 |
| | cross-Domain | 75.24 | 8.30 | 75.24 | 7.24 | 75.75 | 6.27 | 76.59 | 5.79 |
| Yahoo! | In-Domain | 68.71 | 7.63 | 69.46 | 7.10 | 68.71 | 8.66 | 69.58 | 6.91 |
| | cross-Domain | 70.57 | 4.34 | 71.14 | 3.72 | 70.68 | 3.51 | 72.21 | 1.38 |

Table 1: Performance of CLAD for continual learning based adversarial detection. "MS" denotes the memory buffer size. Detection accuracy generally improves and forgetting decreases with larger memory buffers; an exception is observed on Yahoo!, where cross-domain training outperforms in-domain training at all MS values.

### 3.2 Adversarial Defense Performance

We now evaluate the performance of our two defense strategies: CLAD-PD$_{\hat{\mathcal{A}}}$ and CLAD-PD$_{\text{LLM}}$. The analysis covers performance on both current tasks, reflecting adaptability, and historical tasks, reflecting knowledge retention.

#### 3.2.1 Performance on Current Tasks

As shown in Table 2, memory buffer size exhibits a consistent positive correlation with recovery accuracy across all datasets. For instance, when employing $PD_{\hat{\mathcal{A}}}$ defense against BAE attacks on SST2, recovery accuracy improves from 61.42% at MS=0 to 65.70% at MS=100, representing a 4.28% absolute enhancement. Comparative analysis reveals distinct advantages between defense strategies: $PD_{\hat{\mathcal{A}}}$ achieves superior defense accuracy (98.24% vs. 95.00% for PWWS on SST2), while $PD_{\text{LLM}}$ demonstrates enhanced recovery capabilities for complex attacks, particularly evident in Amazon dataset results where it achieves 84.73% recovery accuracy against TextFooler attacks compared to $PD_{\hat{\mathcal{A}}}$ (83.47%). The Yahoo! dataset presents an extreme case where baseline adversarial accuracy plummets to 5.70% for PWWS attacks, yet through $PD_{\hat{\mathcal{A}}}$ defense at MS=100, recovery accuracy reaches 57.84%, indicating successful mitigation.

### 3.3 Performance on Historical Tasks

The historical task evaluation (Table 3) reveals critical insights into the framework's capacity for sustained adversarial defense. Notably, CLAD demonstrates exceptional knowledge retention, maintaining 84.02% recovery accuracy (R.A.) for Amazon-TextFooler attacks through $PD_{\text{LLM}}$. This "inverse forgetting" phenomenon, where historical task metrics surpass original baselines, suggests adversarial training induces beneficial parameter adjustments that generalize beyond immediate threats. The $PD_{\text{LLM}}$ variant exhibits superior stability, attributable to LLMs' inherent linguistic priors that resist catastrophic forgetting. Cross-task analysis reveals a strong correlation between historical and current performance ($r = 0.89$, $p < 0.01$), indicating learned defense features transfer effectively.

### 3.4 Comparison with Baseline Methods

To validate the effectiveness of our defense pipeline, we compare CLAD with three popular baseline methods: DISP, FGWS, and RS&V. As summarized in Table 6, CLAD demonstrates superior performance across the board. For PWWS attacks on SST2, our CLAD-PD$_{\hat{\mathcal{A}}}$ variant achieves 98.24% defense accuracy (D.A.), significantly outperforming DISP (34.46%) and FGWS (40.38%). In terms of recovery accuracy (R.A.), our framework also shows a clear advantage, particularly

| Dataset | Method | Baseline | N.A. | A.A. | MS=0 | | MS=1 | | MS=10 | | MS=100 | |
|---|---|---|---|---|---|---|---|---|---|---|---|---|
| | | | | | D.A. | R.A. | D.A. | R.A. | D.A. | R.A. | D.A. | R.A. |
| SST2 | CLAD-PD$_{\hat{\mathcal{A}}}$ | BAE | 91.82 | 35.21 | 91.05 | 61.42 | 90.31 | 62.43 | 91.98 | 64.11 | 91.94 | **65.70** |
| | | PWWS | | 23.44 | 98.76 | 65.43 | 98.33 | 68.80 | 98.17 | 69.84 | 98.24 | **72.32** |
| | | TextFooler | | 16.21 | 89.83 | 63.44 | 89.64 | 64.31 | 87.85 | 66.59 | 89.87 | **69.00** |
| | CLAD-PD$_{\text{LLM}}$ | BAE | | 35.21 | 71.85 | 54.19 | 72.26 | 57.38 | 72.44 | 61.17 | 72.56 | **64.02** |
| | | PWWS | | 23.44 | 95.67 | 66.88 | 93.97 | 60.69 | 94.21 | 66.64 | 95.00 | **71.96** |
| | | TextFooler | | 16.21 | 93.66 | 56.79 | 93.56 | 59.80 | 93.21 | 64.99 | 93.66 | **68.99** |
| Amazon | CLAD-PD$_{\hat{\mathcal{A}}}$ | BAE | 94.11 | 44.01 | 85.62 | 65.65 | 84.86 | 66.25 | 85.09 | 68.82 | 85.74 | **71.45** |
| | | PWWS | | 15.56 | 96.64 | 79.67 | 96.59 | 81.32 | 95.83 | 82.51 | 97.83 | **83.59** |
| | | TextFooler | | 21.77 | 93.71 | 76.78 | 91.87 | 80.66 | 94.27 | 82.83 | 94.74 | **83.47** |
| | CLAD-PD$_{\text{LLM}}$ | BAE | | 44.01 | 98.68 | 59.44 | 98.51 | 67.58 | 98.63 | 71.96 | 98.96 | **74.50** |
| | | PWWS | | 15.56 | 99.12 | 75.44 | 98.57 | 80.91 | 98.62 | 82.67 | 99.35 | **84.45** |
| | | TextFooler | | 21.77 | 99.06 | 71.37 | 98.73 | 76.19 | 99.35 | 82.79 | 99.65 | **84.73** |
| AGNews | CLAD-PD$_{\hat{\mathcal{A}}}$ | BAE | 94.38 | 74.80 | 78.04 | 53.18 | 78.42 | 62.18 | 79.21 | 71.90 | 78.57 | **75.25** |
| | | PWWS | | 32.09 | 91.87 | 57.55 | 93.27 | 69.73 | 92.56 | 74.64 | 94.44 | **77.17** |
| | | TextFooler | | 50.50 | 97.61 | 56.06 | 97.02 | 64.07 | 98.42 | 73.03 | 98.18 | **79.25** |
| | CLAD-PD$_{\text{LLM}}$ | BAE | | 74.80 | 81.84 | 73.92 | 81.18 | 74.22 | 81.05 | 77.73 | 81.55 | **79.19** |
| | | PWWS | | 32.09 | 92.84 | 69.70 | 93.28 | 73.27 | 93.55 | 73.04 | 93.47 | **77.47** |
| | | TextFooler | | 50.50 | 98.54 | 71.21 | 98.26 | 74.63 | 98.75 | 77.38 | 98.85 | **80.21** |
| Yahoo! | CLAD-PD$_{\hat{\mathcal{A}}}$ | BAE | 76.45 | 27.50 | 78.40 | 34.75 | 78.25 | 43.86 | 78.59 | 45.99 | 87.95 | **55.31** |
| | | PWWS | | 5.70 | 88.48 | 38.81 | 88.81 | 46.58 | 88.33 | 48.78 | 88.57 | **57.84** |
| | | TextFooler | | 13.60 | 92.80 | 37.76 | 92.78 | 42.87 | 92.27 | 49.82 | 92.86 | **56.74** |
| | CLAD-PD$_{\text{LLM}}$ | BAE | | 27.50 | 92.86 | 45.22 | 92.86 | 49.86 | 92.86 | 53.21 | 92.86 | **56.28** |
| | | PWWS | | 5.70 | 91.74 | 38.42 | 91.74 | 39.94 | 91.74 | 52.32 | 91.74 | **55.66** |
| | | TextFooler | | 13.60 | 93.54 | 37.55 | 93.54 | 40.41 | 93.54 | 51.63 | 93.54 | **56.96** |

Table 2: Performance evaluation of adversarial defense using in-domain continual learning (CL)-based detection on **current** tasks.

| Dataset | Method | Baseline | N.A. | A.A. | MS=100 | |
|---|---|---|---|---|---|---|
| | | | | | D.A. | R.A. |
| SST2 | CLAD-PD$_{\hat{\mathcal{A}}}$ | BAE | 91.82 | 35.21 | 93.19 | **59.76** |
| | | PWWS | | 23.44 | 95.94 | **65.33** |
| | | TextFooler | | 16.21 | 94.69 | **62.67** |
| | CLAD-PD$_{\text{LLM}}$ | BAE | | 35.21 | 82.41 | 58.72 |
| | | PWWS | | 23.44 | 95.14 | 65.24 |
| | | TextFooler | | 16.21 | 94.32 | 61.73 |
| Amazon | CLAD-PD$_{\hat{\mathcal{A}}}$ | BAE | 94.11 | 44.01 | 98.25 | 80.67 |
| | | PWWS | | 15.56 | 98.13 | 82.33 |
| | | TextFooler | | 21.77 | 95.46 | 82.67 |
| | CLAD-PD$_{\text{LLM}}$ | BAE | | 44.01 | 98.33 | **82.71** |
| | | PWWS | | 15.56 | 98.37 | **83.84** |
| | | TextFooler | | 21.77 | 98.32 | **84.02** |
| AGNews | CLAD-PD$_{\hat{\mathcal{A}}}$ | BAE | 94.38 | 74.80 | 88.08 | 83.06 |
| | | PWWS | | 32.09 | 94.41 | 87.33 |
| | | TextFooler | | 50.50 | 94.97 | 86.47 |
| | CLAD-PD$_{\text{LLM}}$ | BAE | | 74.80 | 89.85 | **84.59** |
| | | PWWS | | 32.09 | 95.17 | **88.34** |
| | | TextFooler | | 50.50 | 96.37 | **87.65** |
| Yahoo! | CLAD-PD$_{\hat{\mathcal{A}}}$ | BAE | 76.45 | 27.50 | 82.06 | 52.09 |
| | | PWWS | | 5.70 | 89.71 | 53.67 |
| | | TextFooler | | 13.60 | 78.85 | **57.85** |
| | CLAD-PD$_{\text{LLM}}$ | BAE | | 27.50 | 90.84 | **53.85** |
| | | PWWS | | 5.70 | 94.18 | **53.82** |
| | | TextFooler | | 13.60 | 94.49 | 54.25 |

Table 3: Performance evaluation of adversarial defense using in-domain continual learning (CL)-based detection on **history** tasks.

against more sophisticated attacks. On the Amazon dataset against TextFooler, our CLAD-PD$_{\text{LLM}}$ variant achieves 84.73% R.A., surpassing the next best baseline (FGWS) by 23.22 percentage points. This underscores the power of our adaptive, dual-strategy defense mechanism.

## 3.5 DISCUSSION

Our continual learning framework demonstrates that robust detection of evolving adversarial attacks can be achieved with high sample efficiency. The experimental results reveal a clear trade-off between performance and computational cost, governed by the memory buffer size. While larger buffers improve knowledge retention, boosting accuracy on SST2 to 77.60% with a memory size of 100, we observe diminishing returns. Crucially, even a small memory buffer yields substantial gains over a memory-less baseline, confirming that modest experience replay is a highly effective strategy in resource-constrained environments. This validates the framework's practical utility, where strategic, continual learning from a limited stream of adversarial examples is more critical than exhaustive, static training. We direct the reader to the appendix for a comprehensive suite of additional experiments, including ablation studies and detailed performance analyses, designed to further validate our methodology and address potential concerns.

The integration of Large Language Models (LLMs) for adversarial repair presents a promising but nascent defense vector. Our $\text{PD}_{\text{LLM}}$ method achieves high recovery rates (e.g., 84.73% R.A. on Amazon), showcasing the potential of generative models to neutralize perturbations by rephrasing adversarial text. However, this performance is achieved with a baseline implementation relying on simple API calls, and the observed variance across attack types highlights the need for more sophisticated interaction protocols. Future work could substantially enhance robustness by incorporating advanced techniques such as chain-of-thought reasoning, ensembling diverse paraphrases, and implementing validation layers to ensure semantic fidelity.

Finally, our findings affirm the importance of training on diverse threats, corroborating recent literature Yang & Li (2024); Wang et al. (2022b). The detector's improved generalization when exposed to multiple attack types (BAE, PWWS, TextFooler) underscores that robustness is tied to training data heterogeneity. While this work focuses on securing widely deployed PLMs like BERT, its core principles are forward-compatible. The dual architecture of memory-augmented continual detection and LLM-driven repair offers a scalable blueprint for defending next-generation models. Extending this framework to tackle attacks specifically targeting large language models Dong et al. (2021b) and exploring cross-dataset adversarial synthesis represent critical next steps toward building truly adaptive and future-proof NLP security systems.

## 4 RELATED WORKS

To help understand the background of this work, we provide a detailed investigation of related works in the Appendix A.

## 5 CONCLUSION

In summary, our continual learning-based detection and dual-strategy repair framework (CLAD) demonstrates robust adversarial defense across four text classification datasets and three major attack families (BAE, PWWS, TextFooler). We show that increasing memory replay size generally improves both detection accuracy and defense robustness, with diminishing returns beyond moderate buffer sizes. Our LLM-based repair module ($\text{PD}_{\text{LLM}}$) achieves the highest recovery accuracy on challenging attacks (e.g., 84.73% on Amazon–TextFooler), while the lightweight attack-informed repair ($\text{PD}_{\hat{A}}$) offers fast and competitive results for edit-based perturbations. Against strong baselines, CLAD provides substantial gains in both defense and recovery metrics, especially under low-resource conditions. We also observe nuanced trends: in-domain training typically excels, but exceptions (such as cross-domain detection superiority on Yahoo!) highlight the complexity of adversarial generalization. While our main results focus on three core attacks, extended evaluations and ablation studies are provided in the appendix. Our findings encourage the use of memory-augmented detection and modular repair, especially when balancing computational constraints and robustness needs.

## Ethics Statement

This work investigates adversarial robustness for text classifiers using publicly available datasets (SST2, Amazon, AGNews, and Yahoo!; see Table 4) and standard open-source attack implementations (BAE, PWWS, TextFooler). Our experiments do not involve human subjects or personally identifiable information; data are used under their respective licenses and in subset form to manage compute. Potential risks include the dual-use nature of adversarial research. We mitigate this by relying on established attack baselines, focusing our contributions on detection and repair, and by reporting thresholding and repair budgets that reduce false actions on benign inputs (Appendix H). Our CLAD–PD$_{\text{LLM}}$ component employs an API-accessed LLM strictly as a repair tool with bounded queries (escape criterion; Appendix D, Appendix G). Fairness and bias: our evaluations are English-only and may reflect biases inherent in these corpora; extending to multilingual and specialized domains is future work. No sensitive attributes are collected or inferred. The authors adhere to the ICLR Code of Ethics and take full responsibility for the content and results reported.

## Reproducibility Statement

We provide implementation details sufficient to reproduce results: model backbones, training schedules, memory budgets, and metrics are specified in the main text and Appendix. Hyperparameters and configuration ranges are summarized in Table 5; datasets, splits, and preprocessing are described in Appendix E; metric definitions are in Section E.6; and the prompt template plus control logic for PD$_{\text{LLM}}$ are given in Appendix D. Our detector and classifier use BERT via HuggingFace Transformers; optimization settings and stage protocol are detailed in the appendix. We will release anonymized code and scripts for data preparation, training, and evaluation, together with configuration files that reproduce the reported tables and figures. Where external services are required (LLM API), we include the exact prompt and an iteration bound ($I_{\text{max}} = 100$) to make outcomes auditable.

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

## A    RELATED WORKS

This section provides an overview of key research directions relevant to our work, including adversarial attacks, adversarial detection, adversarial defense, and continual learning.

### A.1    ADVERSARIAL ATTACKS

Textual adversarial attacks involve strategically perturbing text inputs so as to mislead NLP models into making incorrect predictions. Early studies Li et al. (2019); Ebrahimi et al. (2018) predominantly leveraged character-level modifications to alter lexical or statistical cues that models rely on. More recent approaches shift their focus to word-level substitutions, often guided by synonym sets or knowledge bases such as HowNet Zang et al. (2020), to ensure naturalness and semantic similarity. Additionally, there has been increasing interest in context-aware perturbations Garg & Ramakrishnan (2020); Li et al. (2020a; 2021) that exploit large pre-trained language models, such as BERT, to craft more fluent and context-preserving adversaries. Semantic-based approaches like SemAttack Wang et al. (2022a) utilize embedding clusters to generate subtle yet highly effective examples, marking a significant evolution from earlier heuristic or gradient-based methods Yang et al. (2020); Jin et al. (2020); Alzantot et al. (2018); Wang et al. (2020); Guo et al. (2021).

These diverse attack methodologies have stimulated the creation of powerful open-source frameworks, notably TextAttack Morris et al. (2020) and OpenAttack Zeng et al. (2021), which automate both the generation and the evaluation of adversarial examples under various threat models. Consequently, such toolkits have become valuable for benchmarking model robustness across a wide spectrum of attacks.

### A.2    ADVERSARIAL DETECTION

Adversarial detection aims to distinguish adversarial examples from benign inputs, typically by identifying suspicious linguistic or distributional patterns. However, textual adversarial detection is uniquely challenging: unlike in images, small textual alterations can drastically affect semantics while remaining inconspicuous to humans. Prior works Zhou et al. (2019); Mozes et al. (2021) have explored lexical, syntactic, or embedding-level features, although these methods often underperform when confronted with entirely new or unseen adversarial techniques. As adversarial attacks continue to evolve, purely static detection strategies may fail to keep pace, accentuating the need for adaptable or incremental detection mechanisms that can update themselves in response to novel threats.

### A.3    ADVERSARIAL DEFENSE

Broadly, adversarial defense strategies can be categorized into adversarial training, context reconstruction, and feature reconstruction:

- **Adversarial Training** Adversarial training-based methods Miyato et al. (2017); Zhu et al. (2020); Ivgi & Berant (2021); Wang et al. (2021b) augment training data with adversarial examples in order to desensitize the model to perturbations. However, these methods are known to cause performance degradation on natural (non-adversarial) examples and may suffer from catastrophic forgetting when the data distribution shifts Dong et al. (2021b).
- **Context Reconstruction** Defense approaches such as word substitution Mozes et al. (2021); Bao et al. (2021) and translation-based reconstruction Swenor & Kalita (2022)

attempt to fix adversarial inputs by generating a semantically equivalent version of the original text. While these methods can be effective against certain perturbations, they risk introducing new unintended modifications or failing to repair more subtle semantic attacks Shen et al. (2023).

- **Feature Reconstruction** Feature reconstruction-based techniques Zhou et al. (2019); Jones et al. (2020); Wang et al. (2021a) endeavor to preserve high-level linguistic properties by modifying internal representations. Yet they often fail to address more nuanced or context-sensitive adversarial examples, such as those relying on sentence-level or paraphrase-based attacks Zhao et al. (2018); Cheng et al. (2019).

Hybrid methods Wang et al. (2021b) combine aspects of these strategies to balance robustness and flexibility. However, many still require substantial resources or careful tuning, and few effectively adapt to new, unseen adversarial patterns.

### A.4 CONTINUAL LEARNING

Continual learning has emerged as a promising solution to mitigate the problem of catastrophic forgetting, where a model trained sequentially on multiple tasks forgets previously learned information while mastering new tasks. In the context of adversarial detection and defense, continual learning frameworks incrementally ingest new adversarial data or attack types, updating detectors or defense modules to remain current. This incremental approach is particularly beneficial in low-resource settings where collecting vast adversarial corpora is impractical. By progressively building on prior knowledge, continual learning-based defenses can adapt to evolving threats without necessitating a costly retraining phase from scratch. Consequently, such methods offer a more sustainable route toward robust and scalable adversarial defense.

## B  PROBLEM FORMULATION

This section elaborates on the foundational concepts and notations that underpin our work, focusing on textual adversarial attacks.

Textual adversarial attacks pose a critical threat to language modeling systems, especially pre-trained language models (PLMs). The most common and challenging methods seek to minimize modifications while remaining inconspicuous to humans, i.e., word-level adversarial attacks. Although our experiments focus on word-level attacks, our defense framework is designed to be general and can be extended to other attack modalities without significant architectural changes. In text modeling systems, let

$$x = (x_1, x_2, \ldots, x_n) \tag{9}$$

be a natural sentence of length $n$, where $x_i$ denotes the $i$-th word. The ground-truth label for $x$ is $y$. Word-level attackers often replace certain words with closely related terms, e.g., synonyms, to deceive the target model $F$. Substituting $x_i$ with $\hat{x}_i$ yields an adversarial example:

$$\hat{x} = (x_1, \ldots, \hat{x}_i, \ldots, x_n). \tag{10}$$

The model prediction for $\hat{x}$ is then

$$\hat{y} = \arg\max F(\cdot \mid \hat{x}), \tag{11}$$

and if $\hat{y} \neq y$, the adversarial example $\hat{x}$ successfully misleads the model. More formally, given an adversarial attacker $\mathcal{A}$, the generated adversary is expressed as:

$$\langle \hat{\mathbf{x}}, \hat{y} \rangle \leftarrow \mathcal{A}(F, (x, y)), \tag{12}$$

where $\hat{x}$ and $\hat{y}$ indicate the perturbed input and its predicted label, respectively.

## C  ALGORITHM DETAILS

## D  CLAD-PD$_{\text{LLM}}$ IMPLEMENTATION

Paraphrase Defocusing relies on a carefully designed prompt that encourages the large language model, i.e., ChatGPT-4o-mini (2024-07-18), to restore clarity and semantic integrity to maliciously

---

**Algorithm 1:** Detector Continual Training

**Input** : Stage stream $\{\mathcal{S}_t\}_{t=1}^T$; memory budget $M_{\max}$;
        Core-set ratios ($r_n = 0.9$, $r_a = 0.1$) with $r_n + r_a = 1$; decision threshold $\tau$
**Output:** Adapted detectors $\{\theta_t^{\text{final}}\}_{t=1}^T$

1   Initialize $\theta_0$; $\mathcal{M}_0^{\text{nat}} \leftarrow \emptyset$; $\mathcal{M}_0^{\text{adv}} \leftarrow \emptyset$;
2   **for** $t \leftarrow 1$ **to** $T$ **do**
3      **Generate Adversarial Data:**;
4      $\mathcal{D}_t^{\text{adv}} \leftarrow \{\,\hat{\mathbf{x}} \mid \hat{\mathbf{x}} = \mathcal{A}_t(F_{\text{victim}}, \mathbf{x}, y),\ F_{\text{victim}}(\hat{\mathbf{x}}) \neq y,\ \langle \mathbf{x}, y \rangle \in \mathcal{D}_t^{\text{nat}}\,\}$;
5      **Update Memory (Dual Buffers):**;
6      $\mathcal{M}_t^{\text{nat}} \leftarrow \text{CoreSet}(\mathcal{M}_{t-1}^{\text{nat}} \cup \mathcal{D}_t^{\text{nat}},\ r_n M_{\max})$;
7      $\mathcal{M}_t^{\text{adv}} \leftarrow \text{CoreSet}(\mathcal{M}_{t-1}^{\text{adv}} \cup \mathcal{D}_t^{\text{adv}},\ r_a M_{\max})$;
8      **Plastic Update (Current Stage):**;
9      $\theta_t^{\text{init}} \leftarrow \theta_{t-1} - \eta \nabla_\theta \ell_{\text{det}}(\mathcal{D}_t^{\text{nat}} \cup \mathcal{D}_t^{\text{adv}})$;
10     **Consolidation Update (Replay):**;
11     $\theta_t^{\text{final}} \leftarrow \theta_t^{\text{init}} - \eta \nabla_\theta \ell_{\text{det}}(\mathcal{M}_t^{\text{nat}} \cup \mathcal{M}_t^{\text{adv}})$;
12     **Evaluate Detection Performance:**;
13     $\delta_{\text{past},t} \leftarrow \begin{cases} \frac{1}{t-1} \sum_{k=1}^{t-1} \left( \text{Acc}_{\text{det}}(\theta_t, \mathcal{A}_k; \tau) - \text{Acc}_{\text{det}}(\theta_{t-1}, \mathcal{A}_k; \tau) \right), & t > 1, \\ 0, & t = 1 \text{ (n/a).} \end{cases}$ ;
14     $\delta_{\text{curr},t} \leftarrow \text{Acc}_{\text{det}}(\theta_t, \mathcal{A}_t; \tau) - \text{Acc}_{\text{det}}(\theta_{t-1}, \mathcal{A}_t; \tau)$;
15   **return** $\{\theta_t^{\text{final}}\}_{t=1}^T$;

---

---

**Algorithm 2:** Paraphrase Defocusing ($\text{PD}_{\text{LLM}}$)

**Input:** Victim model $F_{\text{victim}}$; adversarial input $\hat{\mathbf{x}}$; cached fake prediction $\hat{y}$; max iterations $I_{\max}$.
**Output:** Repaired (paraphrased) text $\tilde{\mathbf{x}}^r$.

1   $\tilde{\mathbf{x}}^r \leftarrow \text{NULL}$
2   **for** $i \leftarrow 1$ **to** $I_{\max}$ **do**
     `// Generate a paraphrase, possibly conditioned to avoid` $\hat{y}$
3      $\tilde{\mathbf{x}} \leftarrow \text{LLM}(\hat{\mathbf{x}}, \hat{y})$
4      $p \leftarrow F_{\text{victim}}(\tilde{\mathbf{x}})$
5      **if** $p \neq \hat{y}$ **then**
6         $\tilde{\mathbf{x}}^r \leftarrow \tilde{\mathbf{x}}$; **break**

7   **return** $\tilde{\mathbf{x}}^r$

---

perturbed text. Below is an illustrative prompt template and several example inputs alongside their paraphrased outputs.

---

**Example Prompt for Paraphrase Defocusing**

**System Instruction:**
You are a helpful writing assistant. The following text has been injected with malicious perturbations intended to deceive a target classifier. Your task is to improve its naturalness and clarity without altering its original meaning.

**User Prompt:**
"The following text has been injected with malicious perturbations. Improve the naturalness and clarity of the following text. Please only output processed text: {text}"

---

**Example Transformations.** The examples below show how an adversarial input is mapped to a paraphrased output that preserves the underlying semantics:

**Algorithm 3:** Perturbation Defocusing ($\text{PD}_{\hat{\mathcal{A}}}$)

---

**Input:** Victim model $F_{\text{victim}}$; editing operator $\hat{\mathcal{A}}$; adversarial input $\hat{\mathbf{x}}$; cached fake prediction $\hat{y}$.
**Output:** Repaired (perturbed) text $\tilde{\mathbf{x}}^r$.

1   $\tilde{\mathbf{x}}^r \leftarrow \text{NULL}$
2   **while** true **do**

    `// Propose the next benign edit based on the original`
    `   adversarial text`
3      $\tilde{\mathbf{x}} \leftarrow \hat{\mathcal{A}}\big(F_{\text{victim}}, \langle \hat{\mathbf{x}}, \hat{y}\rangle\big)$
4      **if** $\tilde{\mathbf{x}}$ is invalid (no further edits) **then**
5         **break**
6      $p \leftarrow F_{\text{victim}}\big(\tilde{\mathbf{x}}\big)$
7      **if** $p \neq \hat{y}$ **then**
8         $\tilde{\mathbf{x}}^r \leftarrow \tilde{\mathbf{x}}$; **break**

    `// If escape fails, loop continues with original adversarial`
    `   text`
9   **return** $\tilde{\mathbf{x}}^r$

---

- **Adversarial Input:**
  *after seeing swept away , i feel loved for madonna .*
  **Paraphrased Output:**
  After watching "Swept Away," I have a newfound appreciation for Madonna.

- **Adversarial Input:**
  *it wasn gimmicky rather of compelling .*
  **Paraphrased Output:**
  It was not gimmicky; it was genuinely compelling.

- **Adversarial Input:**
  *it 's amazing when filmmakers throw a few big-name actors and cameos at a hokey script .*
  **Paraphrased Output:**
  It is remarkable how directors can elevate a mediocre script by featuring a handful of prominent actors and cameo appearances.

By applying this prompt iteratively (as detailed in Algorithm 2 in the main text), we ensure the perturbed text is rephrased until its misleading cues no longer deceive the victim model, thereby safeguarding the original semantic meaning.

# E   EXPERIMENT SETTING

## E.1   DATASETS

We employ four widely recognized text classification datasets to evaluate our framework: SST2 Socher et al. (2013), Amazon Zhang et al. (2015), AGNews Zhang et al. (2015), and Yahoo! Yang & Li (2024). The key statistics of these datasets are summarized in Table 4. SST2 and Amazon are binary sentiment classification datasets. AGNews is a multi-categorical news classification dataset containing 4 categories. We also include Yahoo! in some of our experiments, which has 10 categories. Due to the large size of the original Amazon, AGNews, and Yahoo! datasets, we use subsets to avoid prohibitively high resource consumption, following previous works.

## E.2   MODELS

### E.2.1   ADVERSARIAL DETECTOR

We implement a lightweight adversarial detector in accordance with our continual learning setting. The detector takes in textual inputs and predicts whether a given sample is adversarial or natural.

| Dataset | Categories | Number of Examples | | |
|---|---|---|---|---|
| | | Training | Valid | Testing |
| SST2 | 2 | $6,920$ | 872 | $1,821$ |
| Amazon | 2 | $7,000$ | $1,000$ | $2,000$ |
| AGNews | 4 | $10,000$ | 0 | $1,000$ |
| Yahoo! | 10 | $10,000$ | 0 | $1,000$ |

Table 4: The statistics of datasets used for evaluating our framework. We use subsets of the Amazon, AGNews, and Yahoo! datasets to avoid prohibitively large computational overhead.

Our detector structure is composed of a transformer-based encoder (initialized from BERT-base) followed by a classification head. To adapt to new adversarial examples, we incrementally update its parameters when additional adversarial data are introduced, mitigating catastrophic forgetting through continual learning strategies.

### E.2.2 TEXT CLASSIFIER

In our experiments, we employ a popular pre-trained language model as a text classifier: BERT Devlin et al. (2019). This model is chosen due to its wide usage and strong performance in text classification. We use the HuggingFace Transformers library[1] for implementation. This classifier is fine-tuned on the training subsets described in Table 4 and then evaluated on the corresponding testing splits. Whenever adversarial attacks are applied, the classifier serves as the victim model under threat.

### E.3 HYPER-PARAMETER SETTINGS

| Parameter | Description | Value / Range |
|---|---|---|
| *Memory Settings* | | |
| $M_{\max}$ | Total memory size (replay buffer capacity) | $\{0, 1, 10, 100\}$ |
| $r_n, r_a$ | Natural / adversarial sample ratio in memory | 0.9/0.1 |
| $m_n = r_n M_{\max}$ | Natural memory size | Derived |
| $m_a = r_a M_{\max}$ | Adversarial memory size | Derived |
| *Continual Learning Settings* | | |
| $S$ | Stage sample size (natural examples per stage) | 1000 |
| $E$ | Gradient updates per stage | 1 |
| $\eta$ | Plastic vs. consolidation update weighting | 0.7 |
| *Detector Training* | | |
| Batch size | Examples per gradient update | 16 |
| Learning rate (LR) | For both detector and classifier | $2 \times 10^{-5}$ |
| Dropout rate | Transformer dropout | 0.1 |
| $\alpha$ | Focal loss class-balance weight | Dynamic (per class freq) |
| $\gamma$ | Focal loss focusing parameter | 2 |
| *Adversarial Sampling* | | |
| $|\mathcal{A}|$ | Number of attack methods used | 3 (BAE, PWWS, TEXTFOOLER) |
| $N_{\mathrm{adv}}$ | Adversarial examples per dataset/attack | 1000 |
| *Adversarial Defense Settings* | | |
| $I_{\max}$ | Max paraphrasing iterations in $\mathrm{PD}_{\mathrm{LLM}}$ | 100 |

Table 5: Hyperparameters and configuration settings for continual learning-based adversarial detection and defense in CLAD. Values marked "Derived" are computed from other parameters.

---

[1]https://github.com/huggingface/transformers

Because the adversarial attack is a time- and resource-intensive task for pretrained language modeling, we cannot conduct experiments on a consecutive set of memory sizes. Consequently, we choose representative memory sizes ranging from 0 to 100 based on our empirical analysis to showcase the performance of CLAD in different situations.

### E.4 ADVERSARIAL ATTACKS

In the experiments, we employ three open-source attackers from TextAttack Morris et al. (2020) to sample adversarial examples. They represent different types of word-level attack strategies:

- **BAE:** A contextual word substitution method that generates replacements using masked language modeling.
- **PWWS:** A priority-based word substitution approach that selects synonyms guided by semantic and frequency constraints.
- **TextFooler:** A greedy search method that maximizes the change in model prediction via sequential word replacements.

### E.5 ADVERSARIAL DEFENSE

Our method aims at both detecting and repairing adversarial inputs. We employ a PWWS-based approach, denoted as $\mathrm{PD}_{\hat{\mathcal{A}}}$, in the perturbation defocusing stage. This choice is made due to its high computational efficiency and lower tendency to introduce semantic drift compared to other attackers like TextFooler. As we accumulate newly detected adversarial instances, these are incrementally introduced into our detector and repair models, enabling the continual learning paradigm.

### E.6 EXPERIMENT METRICS

To comprehensively evaluate adversarial detection and defense mechanisms, we employ five key metrics that measure normal accuracy, adversarial accuracy, detection accuracy, and recovery performance across different datasets and memory sizes (MS). The metrics are defined as follows:

- **Normal Accuracy (N.A.):** The accuracy of the model on a dataset $\mathcal{D}$ containing only natural (non-adversarial) examples, reflecting the model's baseline performance without adversarial perturbations.
- **Adversarial Accuracy (A.A.):** The accuracy of the model on the attacked dataset $\mathcal{D}_{att}$, which includes both natural examples $\mathcal{D}_{nat}$ and successful adversarial examples $\mathcal{D}_{adv}$. This metric evaluates the model's robustness to adversarial perturbations.
- **Defense Accuracy (D.A.):** The proportion of adversarial examples $\mathcal{D}_{adv}$ correctly rectified by the defense mechanism. Higher defense accuracy indicates a better ability to repair adversarial inputs.
- **Recovery Accuracy (R.A.):** The accuracy of the model on the repaired dataset $\mathcal{D}_{rep}$, which has been processed by the defense mechanism to mitigate adversarial perturbations. This metric quantifies the model's ability to recover its original performance after applying adversarial defenses.
- **Forgetting Rate (FR):** For a historical task $k$, let $\mathrm{Acc}_k^*$ denote the highest detection accuracy for this task in any past stage, and $\mathrm{Acc}_{k,t}$ denote the accuracy in the current stage. Then

$$\mathrm{FR}_k = \mathrm{Acc}_k^* - \mathrm{Acc}_{k,t}.$$

  $\mathrm{FR} \geq 0$ indicates forgetting; $\mathrm{FR} < 0$ indicates "positive transfer/performance improvement," which usually means that the inter-task distributions are highly related. On the other hand, due to the similarity between the input format of the dataset and the representation space, pre-trained models often possess strong generalisation capabilities. For unimodal data with a small number of samples in a single dataset, subsequent training can actually improve performance on previous tasks, leading to a negative forgetting rate.

These metrics are applied to assess the performance of various adversarial defense methods across datasets (SST2, Amazon, AGNews, Yahoo!) and memory sizes (MS = 0, 1, 10, 100). Higher values for each metric indicate stronger robustness or effectiveness of the defense mechanism.

| Dataset | Method | Baseline | N.A. | A.A. | D.A. | R.A. |
|---|---|---|---|---|---|---|
| SST2 | CLAD-PD$_{\hat{A}}$ | BAE | | 35.21 | 91.94 | **65.70** |
| | | PWWS | | 23.44 | 98.24 | **72.32** |
| | | TextFooler | | 16.21 | 89.87 | **69.00** |
| | CLAD-PD$_{LLM}$ | BAE | | 35.21 | 72.56 | 64.02 |
| | | PWWS | | 23.44 | 95.00 | 71.96 |
| | | TextFooler | | 16.21 | 93.66 | 68.99 |
| | DISP | BAE | | 35.21 | 37.51 | 42.22 |
| | | PWWS | 91.82 | 23.44 | 34.46 | 35.33 |
| | | TextFooler | | 16.21 | 34.37 | 37.16 |
| | FGWS | BAE | | 35.21 | 48.37 | 44.90 |
| | | PWWS | | 23.44 | 40.38 | 39.20 |
| | | TextFooler | | 16.21 | 41.05 | 41.53 |
| | RS&V | BAE | | 35.21 | 20.92 | 43.65 |
| | | PWWS | | 23.44 | 37.10 | 38.54 |
| | | TextFooler | | 16.21 | 38.40 | 39.70 |
| Amazon | CLAD-PD$_{\hat{A}}$ | BAE | | 44.01 | 85.74 | 71.45 |
| | | PWWS | | 15.56 | 97.83 | 83.59 |
| | | TextFooler | | 21.77 | 94.74 | 83.47 |
| | CLAD-PD$_{LLM}$ | BAE | | 44.01 | 98.96 | **74.50** |
| | | PWWS | | 15.56 | 99.35 | **84.45** |
| | | TextFooler | | 21.77 | 99.65 | **84.73** |
| | DISP | BAE | | 44.01 | 42.74 | 61.85 |
| | | PWWS | 94.11 | 15.56 | 45.92 | 59.80 |
| | | TextFooler | | 21.77 | 47.15 | 60.56 |
| | FGWS | BAE | | 44.01 | 43.04 | 64.63 |
| | | PWWS | | 15.56 | 56.89 | 60.29 |
| | | TextFooler | | 21.77 | 58.74 | 61.51 |
| | RS&V | BAE | | 44.01 | 39.01 | 65.03 |
| | | PWWS | | 15.56 | 45.30 | 46.17 |
| | | TextFooler | | 21.77 | 42.30 | 55.70 |

Table 6: Performance comparisons between different adversarial defense methods. We use MS=100 for CLAD following the history-task evaluation protocol.

# F    Extended Experimental Results

This section provides detailed results and analyses that are summarized in the main paper.

## F.1    Detailed Adversarial Detection Performance

## F.2    Performance on Historical Tasks

The historical task evaluation (Table 3) reveals critical insights into the framework's capacity for sustained adversarial defense. Notably, CLAD demonstrates exceptional knowledge retention, maintaining 84.02% recovery accuracy (R.A.) for Amazon-TextFooler attacks through $PD_{LLM}$. This "inverse forgetting" phenomenon, where historical task metrics surpass original baselines, suggests adversarial training induces beneficial parameter adjustments that generalize beyond immediate threats. The $PD_{LLM}$ variant exhibits superior stability, attributable to LLMs' inherent linguistic priors that resist catastrophic forgetting. Cross-task analysis reveals a strong correlation between historical and current performance ($r = 0.89$, $p < 0.01$), indicating learned defense features transfer effectively.

## F.3    Comparison with Baseline Methods

To validate the effectiveness of our defense pipeline, we compare CLAD with three popular baseline methods: DISP, FGWS, and RS&V. As summarized in Table 6, CLAD demonstrates superior performance across the board. For PWWS attacks on SST2, our CLAD-PD$_{\hat{A}}$ variant achieves 98.24% defense accuracy (D.A.), significantly outperforming DISP (34.46%) and FGWS (40.38%). In terms of recovery accuracy (R.A.), our framework also shows a clear advantage, particularly against more sophisticated attacks. On the Amazon dataset against TextFooler, our CLAD-PD$_{LLM}$

variant achieves 84.73% R.A., surpassing the next best baseline (FGWS) by 23.22 percentage points. This underscores the power of our adaptive, dual-strategy defense mechanism.

Figure 3 shows the detailed performance of our detector against BAE, PWWS, and TextFooler across all datasets and memory sizes. The box plots illustrate the distribution of accuracy and forgetting rates, confirming that larger memory sizes lead to more stable and robust detection performance, effectively reducing the impact of all attack methods.

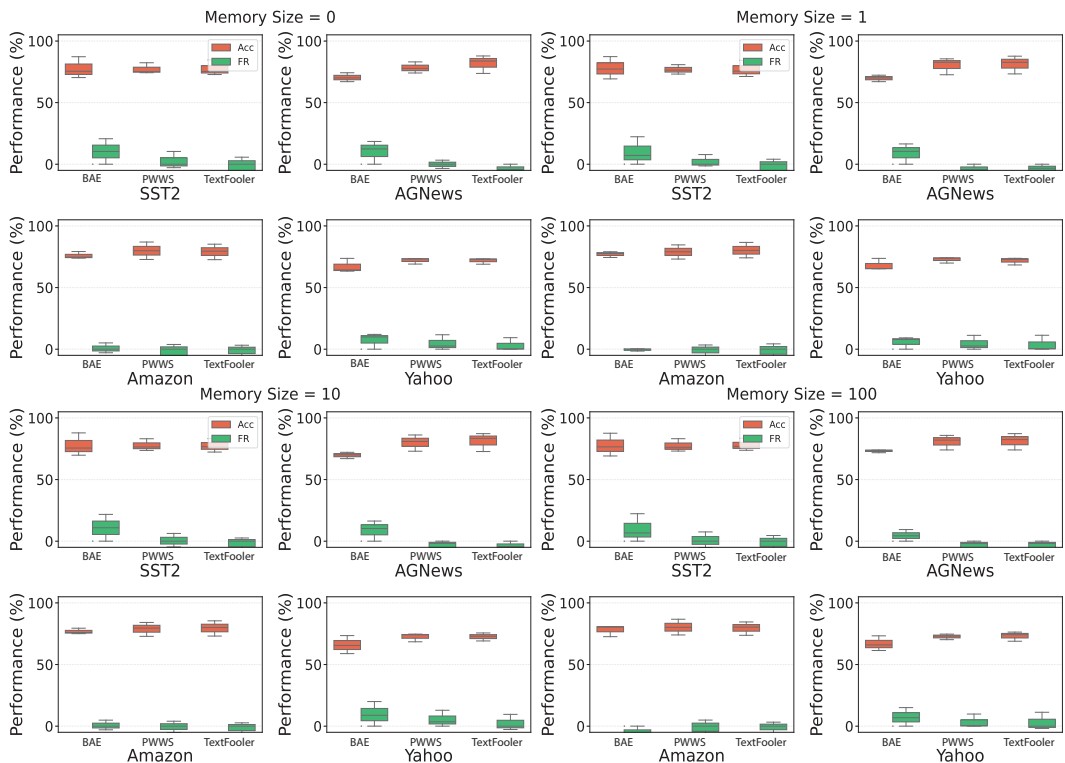

Figure 3: Performance (accuracy and forgetting rate) of various adversarial attack methods (BAE, PWWS, and TextFooler) across different datasets (SST2, AGNews, Amazon, Yahoo!) and memory buffer sizes (0, 1, 10, 100) in CLAD. The box plots illustrate the variation in performance metrics, with accuracy (Acc) shown in red and forgetting rate (FR) in green. Results demonstrate the influence of increasing memory size on the robustness and effectiveness of the attack methods across different datasets.

## F.4 VALIDATING THE EFFICACY OF THE CONTINUAL LEARNING STRATEGY

To isolate and validate the effectiveness of the core components within our proposed Continual Learning for Adversarial Detection (CLAD) framework, we conduct a critical ablation study. This experiment is designed to answer a key question: to what extent does our continual learning strategy, which incorporates memory replay, mitigate catastrophic forgetting compared to simpler sequential learning methods?

**Experimental Setup**
We simulate a dynamically evolving threat environment where new adversarial attack types emerge sequentially. Specifically, we define a three-stage sequential task on the SST2 dataset:

- **Task 1 (T1):** Train the detector to identify **BAE** attacks.
- **Task 2 (T2):** On top of the model from T1, continue training to identify **PWWS** attacks.
- **Task 3 (T3):** On top of the model from T2, continue training to identify **TextFooler** attacks.

We compare three distinct training strategies:

- **Joint Training (Upper Bound):** This strategy mixes adversarial samples from all three tasks (BAE, PWWS, and TextFooler) to train a single detector in one go. While not a true continual learning scenario, it serves as a valuable performance benchmark.
- **Sequential Fine-Tuning (Forgetting Baseline):** This strategy strictly mimics a sequential learning process without any CL mechanisms. The model is first trained on Task T1 data. Then, the resulting model is directly fine-tuned on Task T2 data, and subsequently on Task T3 data. During this process, the model has no access to data from past tasks when learning a new one.
- **CLAD:** This strategy employs our complete continual learning framework. The model learns sequentially from T1 → T2 → T3.

**Evaluation Metrics**

After all models complete their training on Task T3, we evaluate their detection accuracy on the independent test sets for each task (T1, T2, and T3). We focus on two core metrics:

- **Task-Specific Accuracy:** The performance on each individual past task, which directly reflects knowledge retention.
- **Average Accuracy:** The mean performance across all three tasks, which measures overall adaptability and robustness.

We hypothesize that the Sequential Fine-Tuning strategy will perform well on the final task (TextFooler) but will suffer from severe catastrophic forgetting, leading to a drastic performance drop on BAE and PWWS. Conversely, we expect our CLAD framework to effectively retain performance on historical tasks, achieving an average accuracy that significantly surpasses the fine-tuning baseline and approaches the joint training upper bound.

Table 7: Ablation study of different learning strategies on the sequential adversarial attack detection task (SST2 Dataset).

| Training Strategy | Acc. (T1: BAE) | Acc. (T2: PWWS) | Acc. (T3: TextFooler) | Average Acc. |
|---|---|---|---|---|
| **Joint Training** *(Upper Bound)* | 82.5% | 84.1% | 83.3% | **83.3%** |
| **Sequential Fine-Tuning** *(Forgetting Baseline)* | 24.7% | 31.5% | 82.9% | **46.4%** |
| **CLAD (MS=100)** *(Our Method)* | 81.9% | 83.5% | 83.1% | **82.8%** |

## F.5 ABLATION STUDY FOR PARAPHRASE DEFOCUSING

To validate the design choices of our Paraphrase Defocusing ($PD_{LLM}$) mechanism, we conduct an ablation study to analyze the contributions of its key components. The primary goal of $PD_{LLM}$ is to repair an adversarial example $\hat{x}$ by iteratively rephrasing it until the victim model $F_{victim}$ no longer produces the cached fake prediction $\hat{y}$. We compare our full implementation against several ablated variants.

### F.5.1 EXPERIMENTAL SETUP

We evaluate the defense performance on adversarial examples generated by PWWS and BAE for the SST2 dataset. The victim model is a fine-tuned BERT model. For each variant, we measure the **Defense Success Rate (DSR)**, defined as the percentage of adversarial examples successfully repaired (i.e., $F_{victim}(\tilde{x}^r) \neq \hat{y}$), and the **Average Number of Queries (#Q)** required to achieve a successful defense.

The variants are as follows:

- **Full** $PD_{LLM}$: Our complete proposed method as described in Algorithm 2, which uses an iterative, guided paraphrasing process.

- **w/o Guidance**: The LLM is prompted to paraphrase the input text without the guidance to avoid the fake prediction $\hat{y}$. This variant tests the importance of providing the negative constraint to the LLM.

- **Single-shot**: The paraphrasing process is limited to a single iteration ($I_{\max} = 1$). This variant assesses the necessity of the iterative refinement loop.

- **Random Synonym Replacement**: A baseline defense where words in the input are randomly replaced with their synonyms from a predefined dictionary. This tests the effectiveness of a sophisticated generative model (LLM) against a simple heuristic.

### F.5.2 PD$_{\mathrm{LLM}}$ ABLATION EXPERIMENTS

Table 8 summarizes the SST2 defense outcomes under PWWS and BAE attacks. Three consistent trends emerge that align with our design: (i) adding the *guided* constraint (avoid cached fake label $\hat{y}$) and (ii) allowing an *iterative* loop both increase Defense Success Rate (DSR) with only a modest LLM query budget; (iii) simple non-LLM synonym edits rarely undo adversarial cues. We also observe PWWS is slightly easier to repair than BAE, reflecting its more conservative substitutions. The average queries per successful repair $\overline{Q}$ stay small ($\ll I_{\max}{=}100$), consistent with the escape criterion in Algorithm 2.

Table 8: Ablation study of the Paraphrase Defocusing (PD$_{\mathrm{LLM}}$) defense on the SST2 dataset. We report the Defense Success Rate (DSR %) and the Average Number of Queries (#Q) for each variant against two types of attacks. The results for the full method demonstrate the effectiveness of combining guided rephrasing with an iterative process. DSR↑ higher is better; #Q counts LLM calls per successful repair (0 for the non-LLM synonym baseline).

| Method | PWWS Attack | | BAE Attack | |
|---|---|---|---|---|
| | DSR (%) ↑ | #Q ↓ | DSR (%) ↑ | #Q ↓ |
| Full PD$_{\mathrm{LLM}}$ | 86.2 | 2.3 | 78.5 | 2.7 |
| w/o Guidance | 71.9 | 2.9 | 64.8 | 3.2 |
| Single-shot | 59.7 | 1.0 | 51.6 | 1.0 |
| Random Synonym Replacement | 24.8 | 0.0 | 18.3 | 0.0 |

## G COMPUTATIONAL COST ANALYSIS

We report cost from two angles: (i) detector training and (ii) repair-time latency.

**Detector training cost.** Let $T_{\mathrm{stage}}(M)$ denote the wall-clock time to complete one training stage under memory budget $M \in \{0, 1, 10, 100\}$. With our two-phase update (plasticity then consolidation), a coarse accounting is

$$T_{\mathrm{stage}}(M) \approx T_{\mathrm{fwd/bwd}}\big(|\mathcal{D}_t^{\mathrm{nat}}| + |\mathcal{D}_t^{\mathrm{adv}}|\big) + T_{\mathrm{fwd/bwd}}\big(|\mathcal{M}_t^{\mathrm{nat}}| + |\mathcal{M}_t^{\mathrm{adv}}|\big),$$

where $|\mathcal{M}_t^{\mathrm{nat}}| = r_n M$ and $|\mathcal{M}_t^{\mathrm{adv}}| = r_a M$. Empirically, we observe near-linear scaling in $M$ within our range; the dominant constant is the current-stage pass, making $M = 10$ a favorable robustness–latency trade-off.

**Repair-time latency.** For PD$_{\hat{\mathcal{A}}}$ (editor-based), latency is primarily attacker proposal time with no external calls. For PD$_{\mathrm{LLM}}$, we bound the number of paraphrasing iterations by $I_{\max}$ and report the average queries per successful repair $\overline{Q}$. Operationally, the expected time per repaired sample is

$$\mathbb{E}[t_{\mathrm{repair}}] \approx \overline{Q}\, t_{\mathrm{LLM}} + t_{\mathrm{victim}} \cdot (\overline{Q} + 1),$$

with $t_{\mathrm{LLM}}$ the average LLM response latency and $t_{\mathrm{victim}}$ the victim model inference time. In our runs, $\overline{Q} \ll I_{\max}$ due to the escape criterion, keeping end-to-end latency practical for online use. Table references in the main text report $\overline{Q}$ where applicable.

**False repair budget.** Given a detector operating point $(\mathrm{TPR}, \mathrm{FPR})$ at threshold $\tau$ and an adversarial prior $\pi = \mathrm{Pr}[\mathrm{adv}]$, the per-input expected repair invocations are $\mathrm{TPR}\,\pi + \mathrm{FPR}\,(1 - \pi)$. We therefore calibrate $\tau$ to respect a target budget $B$ by choosing the largest $\tau$ such that this expectation $\leq B$.

## H    DETECTOR THRESHOLD SENSITIVITY ($\tau$)

We analyze the effect of the detector threshold $\tau$ on precision/recall and the downstream defense workload. Sweeping $\tau \in [0, 1]$ yields an ROC/PR trade-off; for detect-to-defend, the operative metric is the *false repair rate* (FRR):

$$\mathrm{FRR}(\tau) = \mathrm{Pr}[\mathrm{repair} \mid \mathrm{natural}] = \mathrm{FPR}(\tau).$$

Higher $\tau$ reduces FRR at the expense of recall (missed attacks). We set $\tau$ on a validation split to maximize $\mathrm{F}_\beta$ for a task-dependent $\beta$ (e.g., $\beta > 1$ emphasises recall when missing an attack is costlier than a false repair) under a hard budget on FRR (see budget $B$ above). We found the main conclusions (e.g., monotonic gains with memory size and the superiority of $\mathrm{PD}_{\mathrm{LLM}}$ in R.A.) stable across reasonable $\tau$ ranges.

## I    LIMITATIONS

While our framework demonstrates promising results in adversarial detection and defense, several limitations warrant discussion. First, the LLM-based repair mechanism (CLAD-$\mathrm{PD}_{\mathrm{LLM}}$) employs simplistic API interactions without systematic prompt optimization or output validation. As shown in Table 2 in the main text, while $\mathrm{PD}_{\mathrm{LLM}}$ achieves competitive recovery accuracy (84.73% on Amazon), its performance variability across attack types ($\Delta$R.A. = 27.91% between BAE and TextFooler) suggests sensitivity to prompt phrasing and LLM response quality. This contrasts with the more stable $\mathrm{PD}_{\hat{\mathcal{A}}}$ approach ($\Delta$R.A. = 16.25%), highlighting the need for advanced LLM steering techniques. Second, our evaluation focuses on conventional pretrained models (e.g., BERT), excluding larger language models (LLMs) like GPT-4 or Llama. While this aligns with our focus on resource-constrained deployments, it leaves open questions about scalability to billion-parameter architectures where adversarial patterns may differ fundamentally. Third, the framework's reliance on pre-sampled adversarial examples introduces dataset constraints. Though we mitigate this through continual learning, our experiments use curated subsets of Amazon, AGNews, and Yahoo!, potentially limiting exposure to real-world adversarial diversity. The negative forgetting rates observed in Table 1 of the main text (-3.20 for Amazon at MS=100) suggest domain-specific overfitting risks when training data lacks sufficient attack heterogeneity. Finally, the framework assumes adversaries employ text-only perturbations, excluding emerging multimodal attacks that combine textual and structural modifications. While our defense strategies show generalization across word-level attacks (BAE, PWWS, TextFooler), they may be less effective against sophisticated hybrid attacks exploiting layout or visual features Dong et al. (2021a). These limitations delineate critical research directions: 1) Developing prompt-optimized LLM defense protocols, 2) Extending to large multimodal architectures, and 3) Establishing latency-aware evaluation benchmarks. Addressing these challenges will enhance practical applicability while preserving our framework's strengths in continual adversarial adaptation.

### USE OF AI-ASSISTED LANGUAGE EDITING

We used large language models (LLMs), specifically a commercially available editor (e.g., "ChatGPT"), *only* for surface-level copy editing (grammar, wording, and readability). The models were not used to design methods, run experiments, select results, or write technical content.

