# OpenReview forum: "CLAD: Continual Learning for Robust Adversarial Text Detection and Repair in Resource-Constrained Scenarios"
_ICLR.cc/2026/Conference — ICLR 2026 Conference Withdrawn Submission_

### Official Review · Reviewer_bfyq · 2025-10-30

**Soundness:** 2
**Presentation:** 2
**Contribution:** 1
**Rating:** 2
**Confidence:** 5

**Summary:**

This paper addresses key challenges in textual adversarial defense, including high computational costs, limited generalization, and sensitivity to distribution shifts, by proposing CLAD, a continual learning-based framework for adversarial detection and repair. The method adapts progressively to new attack patterns through a continual learning mechanism while mitigating catastrophic forgetting, demonstrating strong transferability in resource-constrained settings. The experimental design is rigorous, validating the approach on four text classification datasets against three mainstream attack methods.

**Strengths:**

The problem is well-motivated and addresses critical pain points in existing adversarial defense research

**Weaknesses:**

1. Insufficient reference timeliness. The reference list appears outdated, containing no 2025 references and only four 2024 references (based on manual verification, subject to correction).

2. Inadequate baseline comparison. The main text lacks comparative baseline experiments, while the appendix only includes three baseline methods that are relatively obsolete, missing recent state-of-the-art baselines from the past three years.

3. Outdated evaluation methods. All three adversarial methods used for testing predate 2020, failing to demonstrate the approach's effectiveness against contemporary attack algorithms.

**Questions:**

2. Limited experimental scope. The evaluation omits hard-label black-box attack methods, which are more representative of real-world scenarios.

3. Narrow task coverage. We recommend extending validation to translation and generation tasks, as focusing solely on classification significantly constrains the method's applicability.

---

### Official Review · Reviewer_CpzS · 2025-11-01

**Soundness:** 2
**Presentation:** 2
**Contribution:** 2
**Rating:** 4
**Confidence:** 3

**Summary:**

This paper proposes CLAD, a framework for adversarial text detection and repair designed for resource-constrained settings. The framework uses a "detect-to-defend" paradigm. First, a standalone detector, based on BERT, is trained to identify adversarial inputs. If an input is flagged, it is passed to one of two repair modules: "Perturbation Defocusing" method that uses an attacker as an editor, or "Paraphrase Defocusing" method that uses an LLM to rephrase the text. Experiments are conducted on four datasets against three attack types, showing that the CL approach reduces forgetting and that the PD_LLM repair method achieves high recovery accuracy.

**Strengths:**

1. The paper addresses the important and practical challenge of defending against evolving adversarial attacks, especially under the constraints of catastrophic forgetting.

2. The experiments use multiple datasets, attacks, and ablations (memory size, in- vs. cross-domain) and outperform baselines (DISP, FGWS, RS&V).

**Weaknesses:**

1. The individual parts (BERT detector, paraphrase repair) rely on existing techniques, and the attack methods (BAE, PWWS, TextFooler) used are old, limiting the novelty.
2. The method aims for "resource-constrained scenarios"; however, the repair method, PD_{LLM}, relies on iterative API calls to a large LLM (ChatGPT). This introduces significant computational latency, while the latency implications are not quantified.
3. Many improvements (1–3 %) lack significance testing; variance across seeds is not reported. This makes it impossible to determine if these minor improvements are significant or simply measurement noise.
4. All experiments are conducted using BERT as the victim model. While BERT is a standard PLM, adversarial robustness can be architecture-specific. I wonder whether the CLAD framework's effectiveness is general to other model families.
5. The paper's strongest results (e.g., 84.73% R.A. on Amazon-TextFooler) come from the CLAD-PD_{LLM} variant. This variant leverages a powerful, large-scale LLM (ChatGPT). This is then compared against baselines like DISP, FGWS, and RS&V, which are non-LLM-based methods, making the comparisons seem unfair.

**Questions:**

See in weaknesses.

---

### Official Review · Reviewer_BFjm · 2025-11-02

**Soundness:** 1
**Presentation:** 1
**Contribution:** 1
**Rating:** 2
**Confidence:** 4

**Summary:**

The paper proposes a continual-learning framework for adversarial text detection that uses a coreset-style memory to reduce forgetting as new attacks arrive. It then use the LLM-based Paraphrase Defocusing and Perturbation Defocusing to repair attacked inputs.

**Strengths:**

The framework is designed is clean and easy to follow.

**Weaknesses:**

- The main paper is not self-contained; core details on datasets, attack setups, metrics, and buffer sizes are pushed to the appendix, while the main text spends too much space on prior work.

- New concepts (e.g., coreset, memory buffer size) are introduced without definition at first mention, which hurts readability.

- The technical novelty is limited. The main novelty comes in introducing a CoreSet sampling and LLM-based defence, while both were discussed scarcely in the main paper.

- The experimental setting is narrow for 2025: only BERT-base and perturbation-style attacks on sentiment/news tasks, with little discussion of why these tasks are risk-relevant. Given LLMs are being used in defence, why would the downstream classifier still a BERT?

**Questions:**

See weaknesses

---

### Official Review · Reviewer_iLmT · 2025-11-03

**Soundness:** 2
**Presentation:** 2
**Contribution:** 3
**Rating:** 4
**Confidence:** 3

**Summary:**

The paper proposes CLAD, a continual-learning framework for adversarial text detection and repair aimed at low-resource settings. It first trains an adversarial detector incrementally with newly arrived attack types and to pair it with a repair module that restores perturbed inputs. Experimental results on four classification tasks prove that CLAD can effectively defend adversarial attacks for BERT-based classification systems.

**Strengths:**

1. The proposed continual-learning based defend method is novel and reasonable.
2. The proposed method has the potential to be applied on various types of models and tasks.

**Weaknesses:**

1. The formatting of this paper is somehow problematic. For example, the references in the main context are not in brackets, making the content a little bit messy. Also, the related work section (at least its main part) should be placed in the main content rather than in the supplemental material to save pages.
2. Experiments are all on BERT-based models. While I understand that investigating these models are still important, it will be also interesting to see some results on recent LLMs to strengthen the conclusions in the paper.
3. Experiments are limited on classification tasks. Will CLAD has the same effectiveness on generation-based tasks?

**Questions:**

Please see my review above.

---

### Note · Authors · 2025-11-25

**Comment:**

We would like to withdraw this submission from ICLR. After further consideration, we decided to make substantial revisions and plan to resubmit an improved version to another venue. Thank you for your time and consideration.

**Withdrawal Confirmation:**

I have read and agree with the venue's withdrawal policy on behalf of myself and my co-authors.